# Chemical Composition, In Vitro Antitumor Effect, and Toxicity in Zebrafish of the Essential Oil from *Conyza bonariensis* (L.) Cronquist (Asteraceae)

**DOI:** 10.3390/biom13101439

**Published:** 2023-09-24

**Authors:** Rafael Carlos Ferreira, Yuri Mangueira do Nascimento, Paulo Bruno de Araújo Loureiro, Rafael Xavier Martins, Maria Eduarda de Souza Maia, Davi Felipe Farias, Josean Fechine Tavares, Juan Carlos Ramos Gonçalves, Marcelo Sobral da Silva, Marianna Vieira Sobral

**Affiliations:** 1Postgraduate Program in Natural Products and Bioactive Synthetics, Federal University of Paraíba, João Pessoa 58051-970, PB, Brazil; 2Laboratory of Risk Assessment for Novel Technologies (LabRisk), Department of Molecular Biology, Federal University of Paraíba, João Pessoa 58051-970, PB, Brazil

**Keywords:** essential oil, antiproliferative effect, embryotoxicity

## Abstract

The essential oil from *Conyza bonariensis* (Asteraceae) aerial parts (CBEO) was extracted by hydrodistillation in a Clevenger-type apparatus and was characterized by gas chromatography–mass spectrometry. The antitumor potential was evaluated against human tumor cell lines (melanoma, cervical, colorectal, and leukemias), as well as non-tumor keratinocyte lines using the MTT assay. The effect of CBEO on the production of Reactive Oxygen Species (ROS) was evaluated by DCFH-DA assay, and a protection assay using the antioxidant N-acetyl-L-cysteine (NAC) was also performed. Moreover, the CBEO toxicity in the zebrafish model was assessed. The majority of the CBEO compound was (*Z*)-2-lachnophyllum ester (57.24%). The CBEO exhibited selectivity towards SK-MEL-28 melanoma cells (half maximal inhibitory concentration, IC_50_ = 18.65 ± 1.16 µg/mL), and induced a significant increase in ROS production. In addition, the CBEO’s cytotoxicity against SK-MEL-28 cells was reduced after pretreatment with NAC. Furthermore, after 96 h of exposure, 1.5 µg/mL CBEO induced death of all zebrafish embryos. Non-lethal effects were observed after exposure to 0.50–1.25 µg/mL CBEO. Additionally, significant alterations in the activity of enzymes associated with oxidative stress in zebrafish larvae were observed. These results provide evidence that CBEO has a significant in vitro antimelanoma effect by increasing ROS production and moderate embryotoxicity in zebrafish.

## 1. Introduction

Cancer is a relevant public health problem worldwide [1]. In 2020, approximately 19 million people were diagnosed with cancer and about 10 million deaths were recorded [2].

Currently, cancer treatment mainly includes surgery, radiotherapy, chemotherapy, and targeted therapy [2]. Nevertheless, limitations regarding therapeutic success have been observed, including patients relapsing or without an adequate response to therapy [3], in addition to significant adverse effects and chemoresistance [4]. Therefore, the search for new therapeutic opportunities for the treatment of this disease continues to grow globally [5].

Natural products or their derivatives represent more than 60% of the molecules used for cancer treatment [6]. In this context, essential oils (EOs) have been widely investigated for their pharmacological effects, including antimicrobial [7,8], anti-inflammatory [9], and antitumor effects [10,11,12,13,14,15]. EOs are aromatic oily liquids obtained from many parts of plants such as flowers, seeds, leaves, twigs, and roots. EO is composed of numerous volatile constituents such as sesquiterpenes, monoterpenes, aldehydes, alcohols, esters, and ketones [16]. The specific advantage of EOs appears to be in the synergistic effects of their components, as compared to the individual effects of these molecules [17].

Genus *Conyza* (Asteraceae) comprises about 50 species. The secondary metabolites in *Conyza* plants include alkaloids, terpenoids, steroids, phenolic compounds, flavonoids, and tannins. In most *Conyza* species EOs, the major chemical compounds found are limonene and β-farnesene [18]. *Conyza bonariensis* (L.) Cronquist is an invasive plant, native to South America [19]. Several studies show the biological potential of the essential oil from *C. bonariensis* aerial parts as anti-aging [20], antibacterial [21], and antitumor against HeLa (cervical carcinoma), MCF-7 (breast adenocarcinoma), A549 (lung adenocarcinoma), and HepG2 (hepatocellular carcinoma) human tumor cell lines. Additionally, few reports regarding its toxicity were found. Nevertheless, a significant difference in the chemical composition of these EOs was recorded [20,21]. 

Literature data show differences in the chemical composition of EOs even within the same species [22]. Studies on EOs from *C. bonariensis* aerial parts carried out in various countries have revealed different major components, such as (*E*)-β-farnesene [23,24], germacrene D [25], caryophyllene oxide [26], and methyl ester of matricaria [27,28].

Here, we present the chemical characterization of the EO from *C. bonariensis* aerial parts (CBEO) collected in João Pessoa, Paraíba, Brazil. Additionally, we evaluate its antitumor effect in human cell lines (SK-MEL-28, HeLa, HCT-116, HL-60, and K562), and its toxicity on human peripheral blood mononuclear cells (PBMCs) and zebrafish.

## 2. Materials and Methods

### 2.1. Drugs and Reagents 

Dulbecco’s Modified Eagle’s Medium (DMEM) (Sigma-Aldrich^®^; St. Louis, MO, USA), Roswell Park Memorial Institute 1640 (RPMI) medium (Sigma-Aldrich^®^; St. Louis, MO, USA), Histopaque^®^-1077 (Sigma-Aldrich^®^; St. Louis, MO, USA), Buffered phosphate solution (PBS) (Sigma-Aldrich^®^; St. Louis, MO, USA), doxorubicin (DXR) (Sigma-Aldrich^®^; St. Louis, MO, USA), penicillin–streptomycin (Sigma-Aldrich^®^; St. Louis, MO, USA), 2,7-dichlorodihydrofluorescein diacetate (DCFH-DA) (Sigma-Aldrich^®^; St. Louis, MO, USA), N-acetylcysteine (NAC) (Sigma-Aldrich^®^; St. Louis, MO, USA), hydrogen peroxide (H_2_O_2_) (Sigma-Aldrich^®^; St. Louis, MO, USA), 3-(4,5-dimethylthiazol-2-yl)-2,5-diphenyltetrazolium bromide (MTT) (Sigma-Aldrich^®^; St. Louis, MO, USA), sodium chloride (NaCl) (Sigma-Aldrich^®^; St. Louis, MO, USA), potassium chloride (KCl) (Sigma-Aldrich^®^; St. Louis, MO, USA), calcium chloride (CaCl_2_) (Sigma-Aldrich^®^; St. Louis, MO, USA), magnesium sulfate (MgSO_4_) (Sigma-Aldrich^®^; St. Louis, MO, USA), sodium sulfate (Na_2_SO_4_) Sigma-Aldrich^®^; St. Louis, MO, USA), methylene blue Sigma-Aldrich^®^; St. Louis, MO, USA), dimethylsulfoxide (DMSO) (Dinâmica^®^, Indaiatuba, SP, Brazil), Sodium Dodecyl Sulfate (SDS) (Êxodo Científica^®^, Sumaré, SP, Brazil), phytohaemagglutinin (GIBCO^®^, Grand Island, NY, USA), trypsin 0.25% with ethylenediaminetetraacetic acid (EDTA) (GIBCO^®^, Grand Island, NY, USA), Fetal Bovine Serum (FBS) (GIBCO^®^, Grand Island, NY, USA), fish food (Tetra^®^, Melle, Germany), hexane (Biograde^®^, Anápolis, GO, Brazil).

The drugs and reagent solutions were prepared immediately before use.

### 2.2. Human Tumor Cell Lines

The SK-MEL-28 (human melanoma), HeLa (human cervical cancer), HCT-116 (human colon carcinoma), HL-60 (human promyelocytic leukemia), K562 (chronic myeloid leukemia), and HaCat (human immortalized keratinocytes) cell lines were obtained from Rio de Janeiro Cell Bank (BCRJ), Brazil, and cultured in Dulbecco’s Modified Eagle’s Medium (DMEM) (SK-MEL-28, HeLa and HaCaT) or Roswell Park Memorial Institute 1640 (RPMI) (HCT-116, HL-60 and K562) medium supplemented with 10% Fetal Bovine Serum and 1% penicillin–streptomycin at 37 °C with 5% CO_2_.

### 2.3. Human Peripheral Blood Mononuclear Cells (PBMC)

Blood samples were collected from healthy donors in tubes containing EDTA. Human peripheral blood mononuclear cells (PBMCs) were isolated after centrifugation (400× *g*, 20 °C, for 30 min) of the blood samples in tubes containing Histopaque^®^-1077. Subsequently, the interface containing the mononuclear cells was collected and washed with PBS (400× *g*, 10 min, 20 °C). Twenty-four hours before CBEO or DXR treatment, PBMCs were resuspended in RPMI-1640 medium supplemented with 10% FBS, 1% penicillin-streptomycin, and 2% phytohemagglutinin, and cultured in 96-well plates (1 × 10^6^ cells/mL) at 37 °C with 5% CO_2_ [29]. 

### 2.4. Zebrafish Embryos

Zebrafish embryos (AB wild-type strain) with approximately 1 h post-fertilization (hpf) were provided by the Production Unit for Alternative Model Organisms (UniPOM), Federal University of Paraíba (UFPB), João Pessoa, Brazil. The parents were maintained in a recirculation system with regular monitoring of water quality parameters (pH, ammonia, and nitrite levels). The room temperature (26 ± 1 °C) and photoperiod (14:10 light/dark cycle) were controlled. Fish were fed daily with commercial food and freeze-dried spirulina (Fazenda Tamanduá, Patos, PB, Brazil), and were also monitored for abnormal behavior or disease development.

The day before the experiment, zebrafish adults (male-to-female ratio of 2:1) were transferred to a 7 L spawning tank with a bottom mesh and a quick-opening valve for embryo collection. Embryos were collected on the day of the experiment and cultured in adapted embryonic medium E3 (5.0 mM NaCl, 0.17 mM KCl, 0.33 mM CaCl_2_, and 0.33 mM MgSO_4_) containing 0.005% methylene blue. Only spawning with a fertilization rate of ≥90% was used. Viable embryos (showing a normal cleavage pattern and without morphological changes) were selected under an inverted light microscope (Televal 31, Zeiss^®^, Oberkochen, Germany) at 50× magnification.

### 2.5. Botanical Material and Essential Oil Extraction 

*Conyza bonariensis* (L.) Cronquist branches and leaves (1 kg) were collected from the Medicinal Plant Garden, Institute of Research in Drugs and Medicines of Federal University of Paraíba (UFPB), João Pessoa, Paraíba, Brazil (7°08′30.0″ S–34°50′46.7″ W) in September of 2019. An exsiccate of *C. bonariensis* was identified by Prof. Dr. Maria de Fátima Agra and deposited at Herbarium Lauro Pires Xavier-JPB of UFPB, under the number JPB 26391 (registry number SISGEN ABB39C8). 

Essential oil extraction was performed by hydrodistillation in a Clevenger-type apparatus. The samples were crushed and subjected to distillation for 2 h [30]. After extraction, the essential oil was dried with anhydrous sodium sulfate (Na_2_SO_4_), with a yielding (*w*/*w*) of 1.3%.

### 2.6. Essential Oil Analysis

Gas chromatography–mass spectrometry (GC-MS) analysis was performed using a Shimadzu QP-2010 Ultra Quadrupole MS system, operating at 70 eV ionization energy. A capillary column RTX-5MS (30 m × 0.25 mm i.d., 0.25 μM film thickness) was used with Helium as a carrier gas at a flow rate of 3 mL/min with a 1:100 split. The injector and detector temperatures were set at 220 °C and 280 °C, respectively. The column temperature was programmed from 40 °C (isothermal for 1 min) to 220 °C at a rate of 10 °C/min (remaining isothermal for 2 min at 220 °C). Subsequently, the temperature was increased from 220 °C to 280 °C at a rate of 20 °C/min and held isothermally for 5 min at 280 °C. The ions were scanned in scan mode, ranging from *m/z* 50 to 500. The sample solution was prepared in hexane at a dilution of 999:1 (*v*/*v*) and 1 μL was injected into the chromatograph at a flow rate (split) of 1:200.

To calculate the retention indices of compounds, under the same operating conditions, a series of hydrocarbons (C10 to C40) (Sigma-Aldrich^®^) was injected. The retention index for each compound was determined based on a similarity index above 89% estimated by the libraries (Nist. 08 and Wiley 9) used for compound identification. The retention index was calculated using the chromatogram obtained through the Van Den Dool and Kratz equation [31].

### 2.7. Cytotoxicity Assessment of CBEO in Human Cells

The MTT (3-(4,5-dimethylthiazol-2-yl)-2,5-diphenyltetrazolium bromide) assay was performed to evaluate the CBEO cytotoxicity. MTT is a yellow tetrazolium salt that can permeate cell membranes. In viable cells, MTT is converted to insoluble purple formazan crystals, which can be measured spectrophotometrically. The optical density value is proportional to the number of viable cells [32]. The human cell lines SK-MEL-28, HeLa, HCT-116, HL-60, K562, and HaCaT were cultured in DMEM or RPMI medium supplemented with 10% FBS, 100 U/mL penicillin, and 100 μg/mL streptomycin at 37 °C in a humidified atmosphere with 5% CO_2_. The cell suspension was added to 96-well plates (100 µL/well) at a density of 3 × 10^5^ cells/mL (for SK-MEL-28, HeLa, HCT-116, and HaCaT), 5 × 10^5^ cells/mL (for HL-60 and K562), or 1 × 10^6^ cells/mL (PBMC). After culturing for 24 h, cells were incubated with 100 µL of CBEO (HaCaT and human tumor cell lines: 2.34–300 µg/mL; PBMC: 0.15–20 µg/mL) dissolved in DMSO. Doxorubicin (DXR) (molecular weight—MW: 543.52 g/mol) was used as a standard drug. After 72 h, 110 µL of the supernatant was discarded and 10 µL of the 3-(4,5-dimethylthiazol-2-yl)-2,5-diphenyltetrazolium bromide (MTT) solution (5 mg/mL) was added and incubated for another 4 h. The deposited formazan was dissolved with Sodium Dodecyl Sulfate (SDS) (100 mL/well) [33] and the optical densities were measured using a microplate reader (Synergy HT, BioTek^®^, Winooski, VT, USA) at λ = 570 nm and used to calculate the IC_50_ (half-maximal inhibitory concentration). Three independent experiments were performed in triplicate.

The Selectivity Index (SI) was determined from the ratio between the IC_50_ of the non-tumor cell line (HaCaT) and the IC_50_ of the tumor cell line.

### 2.8. Quantification of Reactive Oxygen Species in Human Tumor Cells

Reactive Oxygen Species (ROS) were quantified by the 2,7-dichlorodihydrofluorescein diacetate (DCFH-DA) reagent oxidation method [34]. DCFH-DA is a non-polar and non-fluorescent probe, that can freely cross cell membranes. Intracellular esterases cleave DCFH-DA to DCFH, which is oxidized by ROS to DCF, a highly fluorescent molecule. Therefore, the number of fluorescent cells is proportional to the amount of intracellular ROS [35]. For this experiment, SK-MEL-28 cells were seeded in 24-well plates at a concentration of 2 × 10^5^ cells/mL. After 24 h, cells were exposed to CBEO (20 or 40 µg/mL), DXR (4 µM), or hydrogen peroxide (H_2_O_2_) (500 µM) in the presence of DCFH-DA (10 µM), and incubated for 30 min, 1 h, or 3 h. After the incubation periods, the cells were trypsinized, washed, and resuspended in PBS. The percentage of fluorescent cells was obtained by flow cytometry from 10,000 events acquired at 530 nm fluorescence and 485 nm excitation wavelengths. Three independent experiments were performed in duplicate.

### 2.9. Evaluation of CBEO Cytotoxicity in the Presence or Absence of N-acetylcysteine (NAC)

To assess the involvement of ROS in CBEO cytotoxicity, SK-MEL-28 cells were added to 96-well plates (100 µL/well) at a density of 3 × 10^5^ cells/mL and incubated for 24 h (37 °C, CO_2_ 5%). After this incubation period, cells were incubated for another 3 h (37 °C, CO_2_ 5%) in the presence or absence of 5 µM of N-acetylcysteine (NAC). Subsequently, cells were treated with CBEO (20 or 40 µg/mL, 100 µL per well) or DXR (4 µM, 100 µL per well) and incubated for 72 h in an atmosphere of 5% CO_2_ and 37 °C. Then, the plates were centrifuged, and 110 µL of the supernatant was removed. Then, 10 µL of the MTT solution (5 mg/mL) was added, followed by incubation for 4 h at 37 °C, CO_2_ 5%. The formazan was dissolved with 100 μL of SDS [32] and the optical densities were measured using a microplate reader (Synergy HT, BioTek^®^) at λ = 570 nm. Three independent experiments were performed in triplicate. 

### 2.10. CBEO Toxicity in Zebrafish Model

#### 2.10.1. Acute Toxicity Test Using Zebrafish Embryos

CBEO acute toxicity was determined by Fish Embryo Acute Toxicity (FET) assay. The FET test was conducted independently with CBEO according to OECD’s guideline number 236 [36] with slight modifications. 

Zebrafish embryos with up to 3 hpf of age were exposed to five increasing concentrations (0.5, 0.75, 1.0, 1.25, and 1.5 mg/L) of CBEO. For each concentration tested, a 96-well plate was prepared, containing 20 fertilized eggs (1 embryo per well) exposed to the test sample, and 4 embryos were exposed only to E3 medium (internal controls). Two additional plates containing embryos exposed to E3 medium (negative control) and 0.1% DMSO (solvent control) were also assayed. The volume of liquid in each well at the beginning of the exposure was 0.3 mL. The plates were protected from evaporation by using their own lid and incubating them in a humidified chamber with controlled temperature (26 ± 1 °C) and humidity (70%).

The exposure was performed for 96 h, and the embryos were analyzed daily for lethality endpoints: egg coagulation; lack of somite formation; lack of detachment of the tail-bud from the yolk sac; and lack of heartbeat. In the presence of these endpoints, the embryo/larva was considered dead. 

The number of deaths was used to calculate the survival rate (survival % = number of alive/total organisms × 100). Additionally, non-lethal effects (eye malformation, otolith malformation, mouth malformation, spine malformation, body pigmentation, hatching delay, yolk sac edema, yolk sac deformation, pericardial edema, head edema, blood clotting, and undersize) were also recorded every 24 h. The exposures were under static conditions (without renovation of the test sample or negative and solvent controls). Observations were using a stereomicroscope (50× magnification) and documented with photographs. After 96 h, surviving larvae were euthanized with eugenol and appropriately disposed of. 

The number of deaths and prevalent non-lethal effects (presence in at least three concentrations) was used to calculate the LC_50_ (median lethal concentration) and EC_50_ (median effective concentration) through probit analysis [37]. These values were also used to determine the NOAEL (No Observed Adverse Effect Level) and LOAEL (Lowest Observable Adverse Effect Level).

#### 2.10.2. Oxidative Stress Biomarker Enzymes in Zebrafish Larvae

The FET test was repeated for CBEO under the same conditions described in item 2.8.1., but at this time the embryos were independently exposed to three sublethal concentrations of CBEO (0.12, 0.25, and 0.50 µg/mL) [38]. After 96 h of exposure, the larvae were quickly frozen in 0.1 M sodium phosphate buffer, pH 7.4.

Subsequently, the larvae were macerated using cold NaCl 0.9% 1:9 (*w*/*v*) solution. Homogenates were centrifuged at 10,000× *g* for 10 min at 4 °C, and the resulting supernatants were used for measurement of soluble protein content and enzymatic activity. The activities of lactate dehydrogenase (LDH), glutathione transferase (GST), acetylcholinesterase (AChE), glutathione peroxidase (GPx), and catalase (CAT) enzymes were measured according to Domingues et al. (2010) [39]. Tests were performed in quadruplicate for each enzyme.

### 2.11. Statistical Analysis

Statistical analysis was performed using GraphPad Prism 8.0.2 (Graphpad Software Inc., San Diego, CA, USA). Results are expressed as the mean ± standard error of the mean (SEM). Data statistical analysis was performed using Analysis of Variance (ANOVA), followed by Tukey’s test (*p* < 0.05). The half-maximal inhibitory concentrations (IC_50_) and their 95% confidence intervals (CI 95%) were obtained by non-linear regression analysis. For embryotoxicity assay, the median lethal concentration (LC_50_) values and the median effect concentration (EC_50_) values were calculated by probit regression analysis.

## 3. Results 

### 3.1. (Z)-2-lachnophyllum Ester Was the Major Compound in the Chemical Characterization of CBEO

The analysis of the chemical profile of CBEO led to the identification of 96.95% of its components. The major compound was (Z)-2-lachnophyllum ester (57.24%) (MW: 176.21 g/mol), and the remaining components were monoterpenes and sesquiterpenes (39.71%), as shown in Table 1.

### 3.2. CBEO Induces Cytotoxicity in Human Cell Lines

CBEO induced the least cytotoxicity on the acute promyelocytic leukemia cell line (HL-60), IC_50_ of 32.20 ± 1.10 µg/mL, while the human melanoma cell line (SK-MEL-28) was the most sensitive to the treatment, IC_50_ of 18.65 ± 1.16 µg/mL. Regarding the non-tumor human keratinocyte cell line (HaCaT), CBEO presented IC_50_ of 56.49 ± 1.03, and the standard drug, doxorubicin (DXR), showed high cytotoxicity (IC_50_: 0.28 ± 0.001 µM) after 72 h of treatment. Then, the Selectivity Indices (SI) of CBEO and DXR were determined using the HaCaT healthy skin cell line as a non-tumor cell model. The CBEO showed the highest SI for SK-MEL-28 cells (3.03), as shown in Table 2.

### 3.3. CBEO Induces Less Cytotoxicity in PBMC Cells Than Doxorubicin

CBEO cytotoxicity was also evaluated in human peripheral blood mononuclear cells (PBMC). CBEO induced a concentration-dependent cytotoxicity. The IC_50_ values were 2.68 ± 1.29 (Figure 1A) and 0.06 ± 1.19 µM (Figure 1B) for CBEO and DXR, respectively, after 72 h of treatment.

### 3.4. CBEO Induces Oxidative Stress in SK-MEL-28 Cells

In the dichlorodihydrofluorescein 2′7-diacetate (DCFH-DA) assay, CBEO treatment induced a significant increase in the percentage of fluorescent cells after 30 min (20 µg/mL: 90.32 ± 1.64%; 40 µg/mL: 92.33 ± 1.96%, *p* < 0.05 for both), 1 h (20 µg/mL: 38.36 ± 1.82%; 40 µg/mL: 71.60 ± 1.78%, *p* < 0.05 for both), and 3 h (20 µg/mL: 8.05 ± 0.36%, *p* < 0.05) of treatment, compared to the control (30 min: 8.22 ± 1.26%; 1 h: 2.18 ± 0.16%; and 3 h: 3.15 ± 0.63%). DXR, which was used as a standard drug, induced a significant increase in the percentage of fluorescent cells after 30 min (75.64 ± 6.02% *p* < 0.05) and 3 h (10.64 ± 0.92%, *p* < 0.05) of treatment, compared to the control. As expected, in the group exposed to hydrogen peroxide (H_2_O_2_), there was a significant increase in the percentage of fluorescent cells, compared to the control (30 min: 97.65 ± 0.86%; 1 h: 98.87 ± 0.17%; and 3 h: 99.45 ± 0.23%, *p* < 0.05 for all) (Figure 2).

### 3.5. CBEO Cytotoxicity in SK-MEL-28 Cells Is ROS-Dependent

After 72 h of CBEO treatment, in the absence of N-acetylcysteine (NAC), a significant reduction in cell viability was observed (20 µg/mL: 54.06 ± 3.96%; 40 µg/mL: 20.97 ± 5.60%, *p* < 0.05 for both) compared to the control (100.00 ± 2.47%). NAC pretreatment significantly reduced the CBEO cytotoxic effect, compared to the groups treated only with CBEO at the respective tested concentrations (20 µg/mL in the presence of NAC: 103.3 ± 2.82%; 40 µg/mL in the presence of NAC: 61.63 ± 1.52%, *p* < 0.05 for both) (Figure 3). 

As expected, NAC pretreatment significantly reduced the DXR cytotoxicity (28.45 ± 1.96%, *p* < 0.05) compared to the group treated with DXR in the absence of NAC (47.23 ± 0.81%) (Figure 3).

### 3.6. Embryotoxicity Induced by CBEO in Zebrafish Model

The embryotoxicity assay was performed using CBEO concentrations ranging from 0.5 to 1.5 µg/mL. After 96 h of exposure, all embryos died at the highest concentration tested. However, the two lowest concentrations (0.5 and 0.75 µg/mL) did not induce any mortality (Figure 4).

The non-lethal effects observed included pericardial edema, yolk sac edema, delayed egg hatching, and egg and blood clotting. Additionally, coagulation and absence of heartbeats were the only lethality outcomes observed in embryos and larvae after CBEO treatments (Figure 5).

As observed in Table 3, CBEO presented a median lethal concentration (LC_50_) of 1.20 µg/mL. The Lowest Observed Adverse Effect Level (LOAEL) values were 0.5 µg/mL for delayed hatching and 1.0 µg/mL for yolk sac edema, pericardial edema, and blood clotting. The No Observed Adverse Effect Level (NOAEL) value was 0.75 µg/mL for yolk sac edema, pericardial edema, and blood coagulation. The median effective concentration (EC_50_) for the non-lethal effects found were 0.99 (delayed hatching) and 1.36 (pericardial edema).

Subsequently, the activity of enzymes related to oxidative stress in zebrafish larvae exposed to low concentrations of CBEO (0.12 to 0.50 µg/mL) was studied. The rationale for choosing these concentrations lies in the fact that when conducting tests with sublethal concentrations, we typically observe the endogenous mechanisms of detoxification and neutralization of Reactive Oxygen Species (ROS) in action, rather than observing the final adverse effects such as cell death and necrosis, which are less informative from a mechanistic standpoint.

As shown in Figure 6, there was a significant reduction in acetylcholinesterase (AChE) activity in zebrafish larvae after treatment with CBEO at all concentrations (0.12 µg/mL: 51.27 ± 0.70 µmol/min/mg; 0.25 µg/mL: 64.12 ± 0.27 µmol/min/mg; and 0.50 µg/mL: 46.01 ± 0.87 µmol/min/mg; *p* < 0.05 for all), compared to the control (77.87 ± 1.12 µmol/min/mg). Furthermore, a significant increase in glutathione transferase (GST) activity was observed after treatment with CBEO (0.25 µg/mL: 63.30 ± 0.36 µmol/min/mg; and 0.50 µg/mL: 80.43 ± 0.50 µmol/min/mg; *p* < 0.05 for both) compared to the control (61.56 ± 0.34 µmol/min/mg). For catalase (CAT) activity, we observed a significant increase in the activity of this enzyme in larvae treated with CBEO (0.12 µg/mL: 3.36 ± 0.16 µmol/min/mg; 0.25 µg/mL: 3.73 ± 0.28 µmol/min/mg; and 0.50 µg/mL: 3.71 ± 0.08 µmol/min/mg, *p* < 0.05 for all) compared to the control (2.36 ± 0.06 µmol/min/mg). In addition, treatment of larvae with 0.25 and 0.50 µg/mL of CBEO induced a significant increase in lactate dehydrogenase (LDH) activity (353.6 ± 0.96 µmol/min/mg and 417.4 ± 14.57 µmol/min/mg, respectively; *p* < 0.05 for both) compared to the control (314.00 ± 2.65 µmol/min/mg). For glutathione peroxidase (GPx) activity, we observed a significant increase in the activity of this enzyme in larvae treated with 0.5 µg/mL CBEO (10.70 ± 0.37 µmol/min/mg, *p* < 0.05) compared to the control (9.52 ± 0.10 µmol/min/mg).

## 4. Discussion

Nature represents a significant source of bioactive products. Thus, research around the world has sought to discover and investigate the biological effects of natural products against diseases such as cancer [40]. The present study focuses on elucidating the chemical composition, in vitro antitumor activity, and the embryotoxicity in zebrafish model of the essential oil extracted from *Conyza bonariensis* (L.) aerial parts (CBEO).

The biological properties of essential oils (EOs) are due to their chemical composition [22]. Geographic origin [41] and environmental conditions such as temperature, precipitation, relative humidity, day length, and light intensity [42] influence the biosynthesis and accumulation of natural products [43]. Consequently, variations in the chemical profiles of EOs can occur from plant to plant, even within the same species [22].

Chemical analysis of the CBEO revealed the (*Z*)-2-lachnophyllum ester, an acetylenic compound, as the major compound (57.24%). However, it is worth noting that the chemical profile of the EO from *Conyza bonariensis* (L.) aerial parts can vary depending on the country and regions where the plant grows. Until now, (*E*)-β-farnesene [23], germacrene D [25], caryophyllene oxide [26], limonene [44], 1*H*-indene-3-carboxaldehyde,2,6,7,7a-tetrahydro-1,5-dimethyl [45], and *allo*-aromadendrene [46] have been identified as the major components of the essential oil from *Conyza bonariensis* (L.) aerial parts. Lundgren et al. (2021) [47] showed the characterization of the EO of *C. bonariensis* (L.) cultivated in the Medicinal Plant Garden of the Institute of Research in Drugs and Medicines of the Federal University of Paraíba (UFPB), João Pessoa, Paraíba, Brazil. These authors obtained a different profile chemical from that obtained in our work, with sesquicineole as the major compound. This shows the influence of conditions such as time of harvest (seasonality), mechanical or chemical injuries, genetic factors and evolution, storage, irrigation, herbivory, attack of fungal pathogens, and activity of the pollinators in the production and composition of EOs, as already described in the literature [48,49]. In addition, we have highlighted the presence of other acetylenic compounds in *C. bonariensis* such as methyl ester of matricaria. Specifically, acetylenic compounds constitute a group of molecules common in species of the Asteraceae family and the *Conyza* genus [50,51,52]. Barbosa et al. (2004) [27] obtained different major constituents in the essential oils from distinct parts of this species, such as the methyl ester of matricaria in the roots (74.4%). Furthermore, Mabrouk et al. (2011) [28] observed distinct chemical profiles of the essential oils from *C. bonariensis* cultivated in Tunisia in different seasons. However, the methyl ester of matricaria remained the predominant compound in all of them (63.5–76.4%). 

Finally, the (*Z*)-2-lachnophyllum ester has also been described as the major component (21.2%) of *Conyza bonariensis* (L.) cultivated at the University of Athens, Athens, Greece [53]. Thus, our study presents an EO from *Conyza bonariensis* (L.) aerial parts with unprecedented chemical profile.

Currently, cancer represents a major public health problem worldwide [54], and many efforts are being made in the search for new therapies. The constituents of CBEO include α-pinene, β-pinene, p-cymene, limonene, terpinen-4-ol, caryophyllene oxide, β-sesquiphelandrene, α-humulene, carvacrol, and thymol [55,56,57,58,59,60,61,62,63,64,65,66,67,68,69]. The major compound of CBEO, (*Z*)-2-lachnophyllum ester, exhibited a significant in vitro antitumor effect on the tumor cell lines MDA-MB-231 (human breast carcinoma), MCF-7 (human breast carcinoma), and 5637 (human bladder carcinoma), with half-maximal inhibitory concentration (IC_50_) values ranging from 7.2 to 53.1 µg/mL [70]. Additionally, (*Z*)-2-lachnophyllum ester also possesses antifungal, antioxidant [71], insecticide [72], and nematicidal [73] properties. Considering the advantage of the synergistic effect of EO compounds compared to the biological effects of these constituents individually [17], the in vitro antitumor activity of CBEO was investigated against different human malignant cell lines. Doxorubicin (DXR), a drug widely used in antineoplastic chemotherapy [74] and in several in vitro studies, including the investigation of EO anticancer effects [75,76,77,78,79], was used as a standard drug. Our data show that CBEO induced the greatest cytotoxicity against SK-MEL-28 human melanoma cells. Still, considering the Selectivity Index (SI), we can suggest that this cytotoxic effect was more selective for SK-MEL-28 cells when compared to the effect observed in healthy human skin cells, HaCaT. Cutaneous melanoma is the most relevant malignant tumor among skin cancers, as it is responsible for the majority of deaths [80]. In 2020, approximately 325,000 patients were diagnosed with skin melanoma and approximately 57,000 died from this disease [81]. Thus, novel therapeutic alternatives are needed, and several EOs have been investigated for the treatment of this disease [82] showing effects such as induction of apoptosis by upregulation of Bax and downregulation of Bcl-2 genes [83], cell cycle arrest, and increase in Reactive Oxygen Species (ROS) production [15].

We also analyzed the effect of CBEO against human peripheral blood mononuclear cells (PBMCs). PBMCs, including monocytes and lymphocytes, are frequently isolated for use in preclinical research [84] as well as in studies of new candidates in anticancer therapy [85,86]. The assessment of cytotoxicity in PBMCs is a relevant indicator of human systemic toxicity of natural products [87]. EO effects have been investigated in PBMCs [15,88,89], demonstrating that this experimental model is useful in the investigation of toxicity in human healthy cells. Here, we show that CBEO induced cytotoxicity against PBMCs in a concentration-dependent manner. However, this toxic effect was lower than that observed for the standard drug DXR. Conventional chemotherapy is not selective [90], and thus, damage can also occur in healthy cells, such as blood cells [91,92,93,94]. EOs composed mostly of monoterpenes and sesquiterpenes show cytotoxicity against PBMCs, such as EO from *Duguetia pycnastera* leaves (IC_50_: 21.28 µg/mL) [95], EO from *Satureja khuzistanica* aerial parts (IC_50_: 28.21 µg/mL) [96], and EO from *Xylopia laevigata* leaves (IC_50_: 35.30 µg/mL) [97]. Thus, the greater cytotoxic effect of CBEO on PBMC (IC_50_: 2.68 µg/mL) may be related to the major compound, (*Z*)-2-lachnophyllum ester, which has no report in the literature.

We investigated the effect of CBEO on the redox state of SK-MEL-28 cells. In the 2,7-dichlorodihydrofluorescein diacetate (DCFH-DA) assay, elevated levels of ROS were obtained after CBEO treatment. In addition, pretreatment with N-acetylcysteine (NAC), an antioxidant molecule [98], significantly reduced the CBEO cytotoxicity, corroborating the involvement of ROS in the antitumor effect induced by this EO. It has been reported that increased ROS production is a common event in EO-induced tumor cell death [99]. In fact, oxidative stress induction has been observed in tumor cells after EO treatment [99,100,101]. Tumor cells are known to have high levels of ROS [102] involved in processes such as the induction of cell proliferation and metastasis, inhibition of apoptosis [103], and angiogenesis stimulation [104]. However, excessive intracellular ROS concentration leads to irreparable damage and death of tumor cells [105]. Antineoplastic agents such as doxorubicin, methotrexate, cisplatin, and topotecan induce apoptosis of tumor cells by inducing oxidative stress [106,107,108,109]. Therefore, the increase in ROS production induced by CBEO treatment is a mechanism involved in the antitumor effect of this EO in SK-MEL-28 cells.

The assessment of drug toxicity is a critical process in the development of new drug candidates [110]. In this context, the zebrafish (*Danio rerio*) genome shares 70% similarity with the human genome [111,112]. The zebrafish embryo test is a highly sensitive toxicity test [113], making this experimental model relevant in human health risk assessments [114,115]. Our study provides an assessment of CBEO toxicity in zebrafish embryos and larvae. No mortality of embryos or larvae was observed for the lowest CBEO concentrations (0.50 and 0.75 µg/mL) after 96 h of exposure. However, increasing CBEO concentrations were related to mortality in a concentration-dependent manner, as noted for other EOs [87,116]. In addition, the median lethal concentration (LC_50_) was considered low (1.20 µg/mL); however, similar results related to high toxicity in the zebrafish model were recorded for other EOs from different species, including *Zingiber ottensii* (LC_50_: 1.00 µg/mL) composed mainly of monoterpenes and sesquiterpenes [117], and *Cupressus sempervirens* (LC_50_: 6.60 µg/mL), whose major compound is α-pinene, a monoterpene [111]. To our knowledge, no zebrafish embryotoxicity results were found regarding the (*Z*)-2-lachnophyllum ester, the major compound of CBEO. In addition, this is the first toxicity report of essential oil from *Conyza* species in this experimental model. 

Furthermore, delayed hatching, yolk sac, pericardial edema, and blood clotting were observed in embryos exposed to concentrations of 0.5–1.25 µg/mL. Similarly, the EO from *Leonurus japonicus* aerial parts, rich in phytol and (−)-caryophyllene oxide, induced effects such as yolk sac edema, curved spine, scattered hemorrhages in the edematous yolk sac, incomplete cardiac development, and pericardial edema [118]. Piasecki et al. (2021) [22] observed shortened tails after exposure of embryos to *Cymbopogon nardus* essential oil. In addition, embryos treated with *Cymbopogon winterianus* essential oil showed slightly shortened tails and mild cardiac changes. In addition, embryos treated with the EO from *Cymbopogon citratus* and *Cymbopogon martini* showed slightly slowed development and shortened tails, respectively. All these EOs have monoterpenes as major compounds, such as geraniol and citronellol. At lower concentrations, they all contain limonene (0.7–10%) [22]. Therefore, the significant sensitivity of these organisms to the action of chemical substances is evident, facilitating the search for information on the toxicity of EOs, which contributes to the development of potential medicines.

We also investigated the effect of CBEO on the activity of lactate dehydrogenase (LDH), glutathione transferase (GST), acetylcholinesterase (AChE), glutathione peroxidase (GPx), and catalase (CAT) enzymes in zebrafish larvae. AChE is responsible for hydrolyzing acetylcholine (ACh) [119], a neurotransmitter molecule that plays an important role in the central and peripheral nervous systems [120]. CBEO reduced AChE activity, indicating probable cholinergic system toxicity. Decreased AChE activity is reported with an increase in oxidative stress [121]. Cells need energy to counteract and repair oxidative stress. LDH involves an anaerobic pathway that helps meet energy demand under such conditions. Hence, increased LDH activity can serve as a biomarker for oxidative stress [120]. CBEO treatment led to increased LDH activity, indicating a higher energy requirement in zebrafish larvae exposed to the treatment [29]. CAT is an antioxidant enzyme that converts hydrogen peroxide (H_2_O_2_) into water and oxygen [122]. CAT is a ROS-scavenger that reduces the intracellular concentration of H_2_O_2_ [123]. CBEO induced an increase in CAT activity, suggesting a protective mechanism against a possible toxic oxidative effect [124]. GPx and GST enzymes belong to the family of glutathione-related enzymes present in the body’s antioxidant defense system [120]. Therefore, the increase in GPx and GST activity observed after exposure to CBEO may be associated with increased antioxidant capacity [29]. Taken together, these results indicate a redox imbalance with modulation of the activity of enzymes related to oxidative stress, possibly in an attempt to combat a probable increase in ROS production induced by CBEO.

Our results provide the characterization of an unprecedented EO from *Conyza bonariensis* (L.) that demonstrates significant in vitro antimelanoma activity through the induction of oxidative stress. Regarding the toxicity of CBEO, significant cytotoxicity towards PBMCs was observed. Additionally, an oxidative imbalance appears to be involved in the moderate toxicity observed in the zebrafish model. Therefore, further assays should be conducted to better understand the pharmacological and toxicological effects of this EO. Furthermore, these findings stimulate the isolation of (*Z*)-2-lachnophyllum ester and subsequent in vitro and/or in vivo tests to understand whether the oil’s activity is related to the phytocomplex or the isolated major component.

## Figures and Tables

**Figure 1 biomolecules-13-01439-f001:**
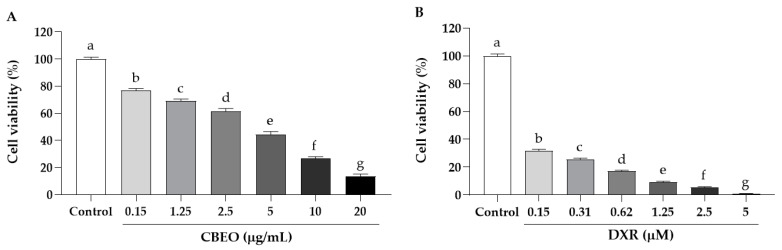
Cytotoxicity of (**A**) essential oil from *Conyza bonariensis* (L.) aerial parts (CBEO) or (**B**) doxorubicin (DXR) against human peripheral blood mononuclear cells (PBMC) after 72 h of treatment. Data obtained from three independent experiments carried out in triplicate and expressed as mean ± standard error of the mean (SEM) analyzed by analysis of variance (ANOVA) followed by Tukey’s test. Different letters denote significant differences among conditions; *p* < 0.05.

**Figure 2 biomolecules-13-01439-f002:**
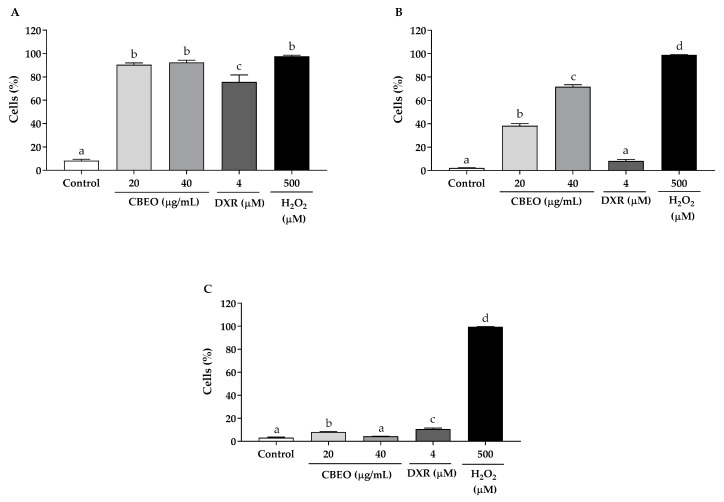
Percentage of 2,7-dichlorofluorescein (DCF) fluorescence cells after (**A**) 30 min, (**B**) 1 h, and (**C**) 3 h of treatment with essential oil from *Conyza bonariensis* (L.) aerial parts (CBEO) (20 or 40 µg/mL) or doxorubicin (DXR) (4 µM). Data obtained from three independent experiments carried out in triplicate were analyzed by analysis of variance (ANOVA) followed by Tukey’s test. Different letters denote significant differences among conditions; H_2_O_2_: hydrogen peroxide; *p* < 0.05.

**Figure 3 biomolecules-13-01439-f003:**
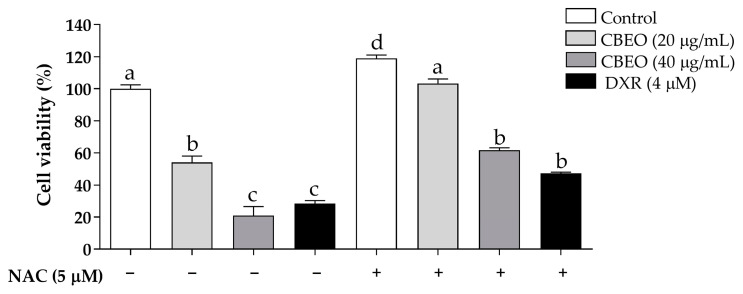
Cytotoxicity of essential oil from *Conyza bonariensis* (L.) aerial parts (CBEO) (20 or 40 µg/mL) or doxorubicin (DXR) (4 µM) in the presence or absence of N-acetylcysteine (NAC) (5 µM) against melanoma cells (SK-MEL-28) after 72 h. Data obtained from three independent experiments carried out in triplicate and expressed as a percentage of cell viability (%) were analyzed by analysis of variance (ANOVA) followed by Tukey’s test. Different letters denote significant differences among conditions; *p* < 0.05.

**Figure 4 biomolecules-13-01439-f004:**
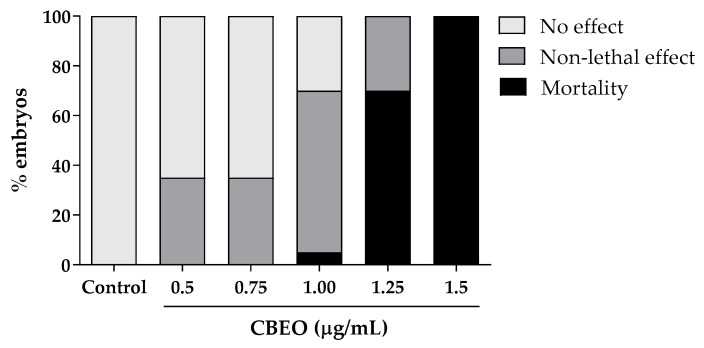
Effects of exposure of zebrafish embryos and larvae to the essential oil from *Conyza bonariensis* (L.) aerial parts (CBEO) after 96 h (*n* = 20 embryos/concentration). No effect: morphological characteristics comparable to control organisms; non-lethal effect: presence of non-lethal endpoints (eye malformation; otolith malformation, mouth malformation, spine malformation, body pigmentation, hatching delay, yolk sac edema, yolk sac deformation, pericardial edema, head edema, blood clotting, and undersize); mortality: presence of lethality outcomes (egg coagulation; lack of somite formation; lack of detachment of the tail-bud from the yolk sac and lack of heartbeat).

**Figure 5 biomolecules-13-01439-f005:**
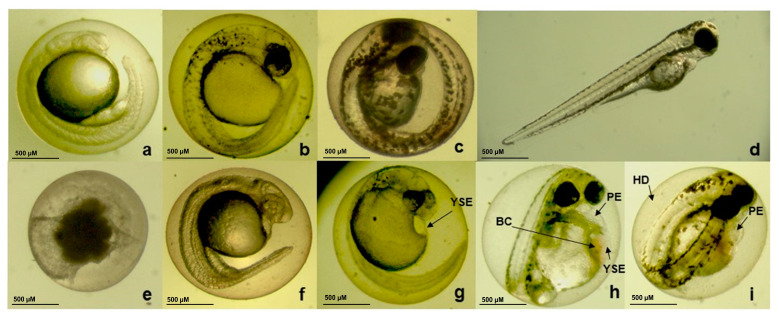
Representative images of lethal and non-lethal effects observed zebrafish embryos and larvae exposed to different concentrations of the essential oil from *Conyza bonariensis* (L.) aerial parts (CBEO) in the embryotoxicity test. In (**a**–**d**), control organisms with normal development after 24, 48, 72 and 96 h, respectively, exposed only to E3 medium; (**e**) embryo after 24 h of exposure to CBEO (1.5 µg/mL) showed egg coagulation; (**f**) embryo after 24 h of exposure to CBEO (0.75 µg/mL) without lethal or non-lethal effects; (**g**) embryo after 48 h of exposure to CBEO (1.0 µg/mL) showed yolk sac edema (YSE); (**h**) embryo after 72 h exposure to CBEO (1.25 µg/mL) showed yolk sac edema (YSE), pericardial edema (PE) and blood clotting (BC); and (**i**) embryo after 96 h of exposure to CBEO (1.25 µg/mL) showed delayed hatching (HD) and pericardial edema (PE). 50× magnification.

**Figure 6 biomolecules-13-01439-f006:**
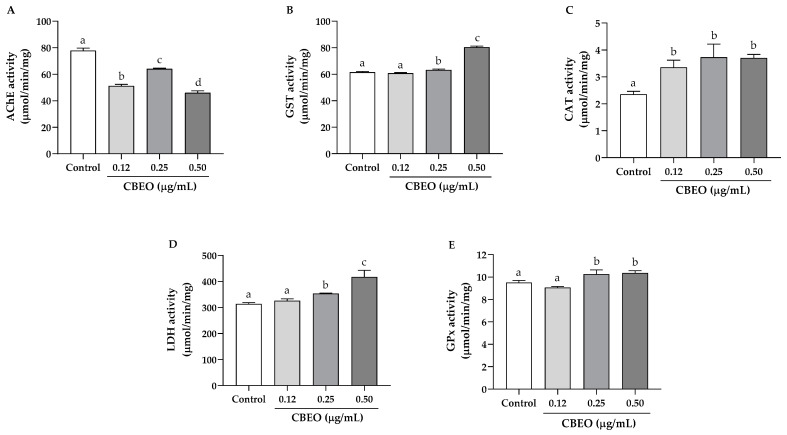
Activity of oxidative stress marker enzymes in zebrafish larvae after 96 h of exposure to different concentrations of essential oil from *Conyza bonariensis* (L.) aerial parts (CBEO). (**A**): acetylcholinesterase (AChE); (**B**): glutathione transferase (GST); (**C**): catalase activity (CAT); (**D**): lactate dehydrogenase (LDH); and (**E**): glutathione peroxidase (GPx) activity. Data are presented as mean ± standard error of the mean (SEM) and were analyzed using one-way ANOVA followed by Tukey’s test. Different letters denote significant differences among conditions; *p* < 0.05.

**Table 1 biomolecules-13-01439-t001:** Identification and quantification of secondary metabolites present in the essential oil from *Conyza bonariensis* (L.) Cronquist aerial parts (CBEO).

Compound	Area (%)	R_t_ ^a^ (min)	RI ^b^ Kovats (Calculated)	RI ^b^ Kovats (Literature)
α-thujene	0.03	6.320	928.792	925.0
(-)-α-pinene	0.66	6.458	937.300	937.0
Sabinene	0.74	7.131	978.792	972.0
β-pinene	1.70	7.209	983.600	976.0
Mircene	0.51	7.376	993.896	993.0
p-cymene	0.08	7.990	1032.198	1024.0
Limonene	14.26	8.062	1036.699	1038.0
(*E*)-β-ocimene	0.04	8.160	1042.826	1047.0
(*Z*)-β-ocimene	1.32	8.333	1053.642	1036.0
Terpinen-4-ol	0.11	10.500	1189.122	1177.0
(*E*,*E*)-2,6-dimethyl-3,5,7-octatriene-2-ol	0.16	10.873	1214.039	1209.2
Thymol	0.24	12.134	1302.998	1297.0
Carvacrol	0.61	12.282	1302.114	1300.0
(*E*)-caryophyllene	4.19	14.046	1442.840	1433.0
(*E*)-α-bergamotene	0.64	14.144	1450.658	1434.0
(*E*)-β-farnesene	0.75	14.316	1464.380	1446.0
(+)-β-funebrene	0.20	14.388	1470.124	1415.0
α-humulene	0.41	14.495	1478.660	1459.0
1-(1,5-dimethyl-4-hexenyl)-4-methylbenzene	0.38	14.730	1497.407	1484.0
Germacrene-D	0.78	14.835	1505.784	1519.0
(*Z*)-2-lachnophyllum ester	57.24	15.105	1527.323	1512.0
β-sesquiphellandrene	7.04	15.265	1540.088	1525.0
(*E*)-nerolidol	0.68	15.678	1573.036	1565.0
Germacrene-B	0.34	15.827	1584.922	1566.0
Spathulenol	1.67	16.065	1604.363	1605.0
Caryophyllene oxide	1.22	16.158	1612.645	1613.0
Isospathulenol	0.41	16.758	1666.073	1666.0
Cadin-4-en-10-ol	0.33	16.937	1682.012	1673.0
Neophytadiene	0.21	18.698	1839.421	1849.0
Total	96.95%			

^a^ R_t_: Retention time; ^b^ RI: Retention index.

**Table 2 biomolecules-13-01439-t002:** Cytotoxicity of the essential oil from *Conyza bonariensis* (L.) aerial parts (CBEO), and doxorubicin (DXR) against human tumor and non-tumor cell lines after 72 h of treatment.

Cell Lines ^a^	IC_50_ ^b^		SI ^c^	
	CBEO (µg/mL)	DXR (µM)	CBEO	DXR
SK-MEL-28	18.65 ± 1.16	3.55 ± 1.67	3.03	0.08
HeLa	30.34 ± 1.08	3.80 ± 1.10	1.86	0.07
HCT-116	31.28 ± 1.16	2.57 ± 0.001	1.81	0.11
HL-60	32.20 ± 1.10	0.22 ± 0.001	1.75	1.27
K562	32.13 ± 1.09	0.71 ± 1.13	1.76	0.39
HaCaT	56.49 ± 1.03	0.28 ± 0.001	-	-

Data obtained from three independent experiments carried out in triplicate and presented as IC_50_ values obtained by nonlinear regression with a 95% confidence interval and expressed as mean ± standard error of the mean (SEM); ^a^ SK-MEL-28: human melanoma cell line; HeLa: human cervical cancer cell line; HCT-116: human colon carcinoma cell line; HL-60: human promyelocytic leukemia cell line; K562: chronic myeloid leukemia cell line; HaCat: human immortalized keratinocytes cell line; ^b^ CI_50_: mean inhibitory concentration; ^c^ SI: selectivity index (IC_50_ non-tumor cell line/IC_50_ tumor cell line).

**Table 3 biomolecules-13-01439-t003:** Effects of exposure to essential oil from *Conyza bonariensis* (L.) aerial parts (CBEO) on developmental parameters of early stages of zebrafish after 96 h.

Embryotoxicological Endpoints	NOAEL ^a^	LOAEL ^b^	EC_50_ ^c^
Eye malformation	n.e. ^†^	n.e. ^†^	n.e. ^†^
Otolith malformation	n.e. ^†^	n.e. ^†^	n.e. ^†^
Mouth malformation	n.e. ^†^	n.e. ^†^	n.e. ^†^
Spine malformation	n.e. ^†^	n.e. ^†^	n.e. ^†^
Body pigmentation	n.e. ^†^	n.e. ^†^	n.e. ^†^
Hatching delay	0.5	n.e. ^†^	0.99 (0.69–1.42) *
Yolk sac edema	1.0	0.75	n.e. ^†^
Pericardial edema	1.0	0.75	1.36 (1.10–1.70) *
Head edema	n.e. ^†^	n.e. ^†^	n.e. ^†^
Blood clotting	1.0	0.75	n.e. ^†^
Undersize	n.e. ^†^	n.e. ^†^	n.e. ^†^
Mortality (LC_50_) ^d^	-	-	1.20 (1.12–1.3) *

^a^ NOAEL: No Observed Adverse Effect Level; ^b^ LOAEL: Lowest Observed Adverse Effect Level; ^c^ EC_50_: median effective concentration; ^d^ LC_50_: median lethal concentration; ^†^ n.e: no effect or less than 20% of embryos affected in the analyzed parameter; * LC_50_ and EC_50_ values are expressed in µg/mL followed by 95% confidence interval (CI) in parentheses.

## Data Availability

The data presented in this study are available in this article.

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
