# Peer review of "Chemical Composition, In Vitro Antitumor Effect, and Toxicity in Zebrafish of the Essential Oil from Conyza bonariensis (L.) Cronquist (Asteraceae)"

_biomolecules, 2023, doi:10.3390/biom13101439_

Round 1

Reviewer 1 Report (Previous Reviewer 2)

I recommend publication of the manuscript.

I have no opinion.

Author Response

Reviewer 2 Report (New Reviewer)

the paper entitled Chemical composition, in vitro antitumor effect, and toxicity in

zebrafish of the essential oil from Conyza bonariensis (L.) Cronquist (Asteraceae)

is focusing on the chemical composition of Conyza bonariensis essential oil (CBEO) (which was previously investigated in other countries) and its potential antitumor effects, and its toxicity in zebrafish.

The Scientific Context of the introduction effectively sets the scientific context by highlighting the global significance of cancer and the need for new therapeutic options. It also discusses the relevance of natural products and essential oils in cancer research.

please cite the previous works that have investigated the essential oil of the same plant such as

Barbosa, L.C., Paula, V.F., Azevedo, A.S., Silva, E.A. and Nascimento, E.A., 2005. Essential oil composition from some plant parts of Conyza bonariensis (L.) Cronquist. Flavour and fragrance journal20(1), pp.39-41.

Mabrouk, S., Elaissi, A., Ben Jannet, H. and Harzallah-Skhiri, F., 2011. Chemical composition of essential oils from leaves, stems, flower heads and roots of Conyza bonariensis L. from Tunisia. Natural product research, 25(1), pp.77-84.

and cite the paper that have investigated  the anticancer effect of the plants such as: Anti-inflammatory and antimitotic effect of the alcoholic extract and chemical composition of the oil from Conyza bonariensis (L.) Cronquist (deer shinbone) leaves

[Efecto antiinflamatorio y antimicótico del extracto alcohólico y composición química del aceite de hojas de Conyza bonariensis (L.) Cronquist (canilla de venado)] Santana et al. Revista Cubana de Plantas Medicinales Volume 16, 2011.

Araujo, L., Moujir, L.M., Rojas, J., Rojas, L., Carmona, J. and Rondón, M., 2013. Chemical Composition and Biological Activity of Conyza Bonariensis Essential Oil Collected in Mérida, Venezuela. Natural Product Communications8(8), p.1934578X1300800838.

and highlight the differences in your and their works in term of cancer cell-lines.

the essential oil of your Conyza bonariensis essential oil contains acetylene derivative called  (Z)-2-lachnophyllum ester (57.24%) present in a very high percentage, why you didn't try to isolate it to perform these biological tests to ensure if the activity was attributed to this compound mainly, it is recommended to highlight the importance of acetylene monoterpene derivative as anticancer agent, this class of compound is common in the essential oil Conyza bonariensis as it was isolated in other samples such as matricaria ester isolated form a Tunisian and Greek samples.

I think you paper needs extensive revision.

Round 2

Reviewer 2 Report (New Reviewer)

Now the article is deeply improved. 

This manuscript is a resubmission of an earlier submission. The following is a list of the peer review reports and author responses from that submission.

Round 1

Reviewer 1 Report

The authors present a chemical and biological characterization of the essential oil from Conyza bonariensis, extracted from branches and leaves collected from the Medicinal Plant Garden, Institute of Research in Drugs and Medicines of the Federal University of Paraíba (UFPB), João Pessoa, Paraíba, Brazil, a sample which was deposited at Herbarium Lauro Pires Xavier-JPB of UFPB, under number JPB 26391 (registry number SISGEN ABB39C8).

The chemical composition of the specimen from C. bonariensis that was deposited at Herbarium Lauro 126 Pires Xavier-JPB of UFPB, under number JPB 26391 (registry number SISGEN ABB39C8), was described in 2021 by Lundgren and co-authors (https://doi.org/10.1111/jam.15244). Why is this work not cited in the manuscript? The results obtained by Ferreira et al. and presented in the manuscript are very different from the ones published by Lundgren et al. l. Why is this not discussed?

For the FET test, what was the volume of liquid in each well at the start of the experiment? Did authors protect plates from evaporation? If not, how did the concentration of the CBEO change with evaporation?

What is known about the properties of (Z)-2-lachnophyllum ester? How it contributes to the in vitro and in vivo effects observed?  

The description of the results from the in vivo experiments needs improvement. Kaplan-Meier survival plot needs to be added. Figure 3 needs to be modified and extended (supplementary Figs). In the title of Fig3, the authors stated that images are representative, but for which group? Judging from Fig 2, fish were divided into 1-3 groups.

Having different concentrations and different time points on one panel is very confusing. It would be beneficial to prepare separate panels for each CBEO concentration so the reader can see the effect of CBEO at different time points. A scale bar should be added!

In Table 3, the authors listed several parameters that were not affected when zebrafish embryos were exposed to CBEO, yet this statement is not supported by any morphometric data and hence cannot be considered valid! Without proper measurement, even a 20% difference in the body length or size of the eye can be easily missed. Are all fish with non-lethal changes affected in a similar way? What do you mean by head enema? 

Minor points:

The source of chemicals/kits used in this study should be given. 

Fish in Fig 3d looks more like 72h- and not 94 hours old.

Which component of CBEO is known / or has been predicted to have antitumor properties?

Lines 157-158: If HL-60 and K562 cell lines were seeded at the same density as SK-MEL-28, HeLA and HCT-116, then it would be more logical to mention them as one group. At what density were HaCAT cells seeded? What was the volume of media in each well? 

Lines 161-163: Is it so that the medium was removed and cells were exposed directly to 5mg/ml MTT solution? Usually, 10ul of the 5mg/ml of MTT stock solution (aqueous or in PBS) is added to wells with cell-culture medium, and the final concentration of MTT is 0.2-0,5mg/ml. 

As the effect of CBEO (concentration is given in ug/ml) is compared with DXR (concentration in uM) it would be nice if at some point MW of DXR (and maybe of the (Z)-2-lachnophyllum ester) could be mentioned.

The manuscript is supposed to be followed with original images, yet the attached figures are not of that type.

Reviewer 2 Report

This paper introduces the chemical composition, anti-tumor effect in vitro and toxicity to zebrafish embryos of the Conyza bonariensis (L.) Cronquist (Asteraceae) essential oil. But all in vitro biological studies were performed based on the crude extracts, which could not provide very valid proof for its exerting biological functions. The following are our specific comments for this manuscript:

1.The authors should briefly explain why Doxorubicin was used as control. Doxorubicin is known to have strong cytotoxic effects.( Line 160 )

2.The x-axis title of Figure 2 is presented well. ( Line 273 )

3.When studying the activity of enzymes related to oxidative stress in zebrafish larvae, it is more helpful for readers to understand the research method by explaining the concentration selection ( 0.12-0.50 μg / mL ). ( Line 320-321 )

4.This paper assumed that the main components of the essential oil from Conyza bonariensis aerial parts, like (E)-β-farnesene, caryophyllene oxide, germacrene, may change due to environmental factors. Is it possible for the main components under other conditions to be changed? This will affect the biological effects. ( Line 354-368 )

5. It is better to explain the strong cytotoxic effect of CBEO on PBMC associated with the main compound (Z)-2-Lachnophyllum ester. ( Line 396-397 )

The quality of English is fine.